# Multi-Omics Analysis Reveals That Anthocyanin Degradation and Phytohormone Changes Regulate Red Color Fading in Rapeseed (*Brassica napus* L.) Petals

**DOI:** 10.3390/ijms25052577

**Published:** 2024-02-23

**Authors:** Lan Huang, Baogang Lin, Pengfei Hao, Kaige Yi, Xi Li, Shuijin Hua

**Affiliations:** 1College of Advanced Agricultural Sciences, Zhejiang A & F University, Hangzhou 311300, China; huang975885829@163.com; 2Institute of Crop and Nuclear Technology Utilization, Zhejiang Academy of Agricultural Sciences, Hangzhou 310021, China; linbg@mail.zaas.ac.cn (B.L.); 11816004@zju.edu.cn (P.H.); kaige_y0113@163.com (K.Y.); 202200101021@stu.xza.edu.cn (X.L.)

**Keywords:** anthocyanin, *Brassica napus* L., fading, metabolomics, petal, phytohormone, transcriptomics

## Abstract

Flower color is an important trait for the ornamental value of colored rapeseed (*Brassica napus* L.), as the plant is becoming more popular. However, the color fading of red petals of rapeseed is a problem for its utilization. Unfortunately, the mechanism for the process of color fading in rapeseed is unknown. In the current study, a red flower line, Zhehuhong, was used as plant material to analyze the alterations in its morphological and physiological characteristics, including pigment and phytohormone content, 2 d before flowering (T1), at flowering (T2), and 2 d after flowering (T3). Further, metabolomics and transcriptomics analyses were also performed to reveal the molecular regulation of petal fading. The results show that epidermal cells changed from spherical and tightly arranged to totally collapsed from T1 to T3, according to both paraffin section and scanning electron microscope observation. The pH value and all pigment content except flavonoids decreased significantly during petal fading. The anthocyanin content was reduced by 60.3% at T3 compared to T1. The content of three phytohormones, 1-aminocyclopropanecarboxylic acid, melatonin, and salicylic acid, increased significantly by 2.2, 1.1, and 30.3 times, respectively, from T1 to T3. However, auxin, abscisic acid, and jasmonic acid content decreased from T1 to T3. The result of metabolomics analysis shows that the content of six detected anthocyanin components (cyanidin, peonidin, pelargonidin, delphinidin, petunidin, and malvidin) and their derivatives mainly exhibited a decreasing trend, which was in accordance with the trend of decreasing anthocyanin. Transcriptomics analysis showed downregulation of genes involved in flavonol, flavonoid, and anthocyanin biosynthesis. Furthermore, genes regulating anthocyanin biosynthesis were preferentially expressed at early stages, indicating that the degradation of anthocyanin is the main issue during color fading. The corresponding gene-encoding phytohormone biosynthesis and signaling, JASMONATE-ZIM-DOMAIN PROTEIN, was deactivated to repress anthocyanin biosynthesis, resulting in fading petal color. The results clearly suggest that anthocyanin degradation and phytohormone regulation play essential roles in petal color fading in rapeseed, which is a useful insight for the breeding of colored rapeseed.

## 1. Introduction

Rapeseed (*Brassica napus* L.) is an important oil crop in China, with a planting area of about 1.5 billion hectares, of which about 80% is distributed in the Yangtze River Basin. Rapeseed has multiple uses in many areas. For example, rapeseed oil is traditionally used as a source of edible vegetable oil for humans. Rapeseed plants in both the vegetative and reproductive stages are also used as feed to support the livestock industry in China. Young stems of rapeseed (10 to 15 cm stalks from the top down, with a plant height of about 40 cm, containing buds, stems, and leaves) are prepared as a high-quality vegetable for people to eat. In addition, rapeseed is also a good ornamental crop. In China, many rapeseed flowers bloom in the winter rapeseed areas of the Yangtze River Basin and the Huanghuai region in March and in spring rapeseed areas of the northwest region, including Gansu, Qinghai, Inner Mongolia Autonomous Region, and Tibet Autonomous Region, and in July every year, providing a beautiful landscape that attracts many tourists [1].

Traditionally, the color of rapeseed petals is yellow. Over time, researchers obtained some mutants with white petals. In recent years, a number of researchers, especially in China, have reported obtaining rapeseed with colors such as red, orange, and purple [2,3,4]. As the petal color of rapeseed has been enriched, researchers have become interested in identifying the main pigment components in different colored petals, the key genes encoding the different colors, and the regulatory mechanisms controlling pigment deposition in petals. For example, epicatechin, quercetin, isorhamnetin derivatives, and cyanidin derivatives were found to be the main pigment components of red rapeseed petals, while kaempferol derivatives were found to be mainly distributed in yellow and white petals, as identified by ultra-high-performance liquid chromatography–HESI–mass spectrometry (UPLC-HESI-MS/MS) [5]. However, other researchers found that the main components of red rapeseed are glycosylated anthocyanins (peonidin-3-O-glucoside, cyanidin-3-O-galactoside, and cyanidin-3-O-glucoside) and kaempferol derivatives [6]. One possible reason for this difference is the different study materials used by researchers, as the color definition of “red” is not consistent. In addition to research on the components of pigment in colored rapeseed petals, research on gene isolation and the mechanism regulating petal pigmentation has also been performed [7]. The carotenoid cleavage dioxygenase 4 (*BnaC3.CCD4*) gene, which was isolated using the positional cloning method, is mainly expressed in white rapeseed petals [8]. In orange petals, two isolated genes, *BnPC1* and *BnPC2*, were found to be responsible for pigment deposition [9,10]. In another study, the change of petal color from yellow to orange was reported to be due to the loss of *BnaC09.ZEP* and deletion of 1695-bp in *BnaA09.ZEP*, revealed by map-based cloning [3]. Recently, Ye et al. (2022) found that *BnaA07.PAP2* was responsible for anthocyanin-based coloration in petals by performing map-based cloning of apricot and pink flowers [11]. In our study, the results show that *BnaA03.ANS* may partly inhibit anthocyanin biosynthesis in red petals of rapeseed under RNA interference, resulting in the weakening of the red color [12]. The above studies enrich our understanding of the pigmentation and regulation mechanism of colored rapeseed petals. However, we found that the red color of rapeseed petals underwent a significant fading phenomenon from before flowering to after flowering, which seriously affects the ornamental effect of colored rapeseed. Unfortunately, the mechanism of this phenomenon in rapeseed has not been reported.

Anthocyanins are the main pigment components of red rapeseed petals [13], so the fading phenomenon may be related to their stability. Anthocyanins are compounds that are susceptible to change, even degradation, under different environmental conditions, such as temperature and light, and their development is regulated by factors such as the pH of cells and the phytohormone content [14,15]. As a result, the color in plant tissues changes. Temperature has an important effect on tissue coloration as well. In general, low temperature promotes the accumulation of anthocyanins, whereas high temperature easily leads to decreasing anthocyanin content [16,17]. Light is another important factor affecting the synthesis and accumulation of anthocyanins in different plant species. Some researchers have reported that strong light can increase the anthocyanin content, whereas anthocyanin synthesis was inhibited under dark or low light conditions, resulting in white or lightly colored flowers [18,19]. In addition to these environmental cues, phytohormones can also regulate anthocyanin synthesis. In most plant species, exogenous jasmonic acid, salicylic acid, abscisic acid, melatonin, and ethylene can promote the accumulation of anthocyanins [20,21], and gibberellin has an inhibitory effect on anthocyanin accumulation [22]. Thus, these examples suggest that the regulation of petal fading in rapeseed might be very complicated.

Petal color fading does not occur only in rapeseed. In *Malus halliana* flowers, increased methylation activity in the R2 and R8 regions was found to suppress the expression of *MhMYB10*, changing the petal color from red to light pink [23]. The change of lotus petals from red to white is the result of the co-regulation of anthocyanin degradation genes and pH [24]. In peony petals, the change of color from coral to pale yellow was found to be due to a decrease in anthocyanins [25]. The above examples indicate that the fading phenomenon of petals is mainly related to decreased anthocyanin content. However, the reason why the color of rapeseed petals fades during flowering and decay remains unknown.

The objective of the current study was to clarify how the pigment of petals changes during color fading and the relationship between pigment content and regulatory factors such as pH and phytohormone content from pre-flowering to post-flowering. Therefore, multiple analyses were conducted at the morphological to the physiological level, as well as metabolomics and transcriptomics, to gain new insights into the mechanism of petal fading for further breeding of color-stable rapeseed varieties.

## 2. Results

### 2.1. Morphological Changes of Red Rapeseed Petals at Different Developmental Stages during Color Fading

The color of Zhehuhong flower petals is red; however, there are obvious differences in the degree of redness during petal color fading (Figure 1a). Stereomicroscopy showed that the deepest color was raspberry at 2 days before flowering (T1). The degree of redness substantially decreased from flowering (T2) to 2 days after flowering (T3), when the color of the petals was obviously faded to beige red (Figure 1a).

In order to reveal the cytological changes during petal color fading in Zhehuhong, paraffin sections were analyzed (Figure 1b–g). The results show that there were differences in the morphology of epidermal cells, parenchymatous cells, and vascular bundles during petal color fading. At T1, the epidermal cells were oval-shaped and tightly arranged, and the cellular size of the lower epidermis was much larger than that of the upper epidermis. The parenchymatous cells had a small size but were tightly arranged and more numerous. The size of vascular bundles was large (Figure 1b,e). At T2, both upper and lower epidermal cells were shriveled, and the cells became rod-shaped. The parenchymatous cells were partly merged and became much larger. The size of vascular bundles was reduced (Figure 1c,f). At T3, the epidermal cells were further shriveled and irregularly arranged, and the parenchymatous cells and vascular bundles had collapsed (Figure 1d,g).

Scanning electronic microscopy showed that the petal cells were spheroid-shaped and tightly arranged (Figure 1h,k). However, at T2 and T3, the cells were wrinkled (Figure 1i,j,l,m).

### 2.2. Physiological Changes of Red Rapeseed Petals in Different Developmental Stages during Color Fading

#### 2.2.1. Reduced pH and Pigments during the Color Fading of Red Rapeseed Petals

The pH value of the red petals of Zhehuhong showed a downward trend in different developmental stages, with all having weak acidity (Figure 2a). The pH of red petals was the highest at 2 days before flowering, reaching 6.49, while the lowest was at 2 days after flowering, reaching 5.23, or a decrease of 19% (Figure 2a).

All measured pigment content except flavonoids in the red petals of Zhehuhong showed a significantly decreasing trend from T1 to T3 (Figure 2b–g). The contents of tannins, proanthocyanidins, flavonoids, carotenoids, total phenols, and anthocyanins were the lowest at 2 days after flowering, decreasing by 24.85%, 45.64%, 17.93%, 53.48%, 49.30%, and 60.31%, respectively, compared to 2 days before flowering (Figure 2b–g).

#### 2.2.2. Changes of Phytohormone Content in Red Rapeseed Petals at Different Developmental Stages

The phytohormone content during petal color fading was also determined. Among the 28 selected compounds, 15 were not detected at any stage: 3-indolebutyric acid, N6-isopentenyladenine, trans-zeatin, cis-zeatin, dihydrozeatin, brassinolide, strigolactone, dihydrojasmonic acid, gibberellin A1, gibberellin A4, gibberellin A7, methylsalicylate, trans-indole-3-acrylic acid, 3-indole propionic acid, and doxifluridine. Among the identified phytohormones, the changes could be classified into five categories: increasing, decreasing, first increasing and then decreasing, first decreasing and then increasing, and no significant change (Figure 3). In the increasing category, 1-aminocyclopropanecarboxylic acid (ACC) and salicylic acid (SA) content increased significantly from T1 to T3, by 2.2 and 30.3 times, respectively (Figure 3). In the decreasing category, methyl indole-3-acetate, kinetin (KT), isopentenyl adenosine, indole-3-carboxaldehyde, (±)-jasmonic acid (JA), and abscisic acid (ABA) content decreased significantly from T1 to T3, with decreases of 53.4%, 84.7%, 83.1%, 69.1%, 92.5%, and 93.0% (Figure 3). In the first increasing and then decreasing category, indole-3-acetic acid (IAA) and methyl jasmonate content increased from T1 to T2 and then decreased at T3. The content of indole-3-acetic acid was 98.4% higher at T2 than at T1. Methyl jasmonate was not detected at T1 and increased by 18.2 mg g^−1^ fresh weight. The content of trans-zeatin-riboside decreased significantly from T1 to T2. Although the content at T3 significantly increased compared to T2, it decreased by 91.8% and 85.4%. Melatonin and gibberellin A3 (GA3) content did not change significantly from T1/T2 to T3 (Figure 3).

### 2.3. Identification of the Differential Accumulation of Metabolites in Zhehuhong Petals during Developmental Stages

To further reveal the different metabolites during petal color fading in Zhehuhong, metabolomics analysis was performed. The results show that there were 137, 103, and 169 distinctive metabolites at T1 vs. T2, T2 vs. T3, and T1 vs. T3, respectively (Figure 4a). Among the identified metabolites, 1414 metabolites were upregulated and 1018 metabolites were downregulated at T1 compared to T2 (Figure 4b). As petal color faded further (T2 to T3), 1015 metabolites were upregulated and 1007 were down regulated (Figure 4c).

After further classification of these metabolites, it was found that the amounts of secondary metabolites were the highest, mainly related to anthocyanin, flavone and flavonol, flavonoid, isoquinoline alkaloid, and phenylpropanoid biosynthesis for T1 compared to T2 and T2 compared to T3 (Figure 4d,e). The results of pathway enrichment analysis show that the most abundant pathway was isoquinoline alkaloid biosynthesis at T1 compared to T2 (Figure 4f). However, isoquinoline alkaloid, flavonoid, and anthocyanin biosynthesis were found to be relatively abundant among the detected pathways (Figure 4g).

In order to further reveal the alteration of components in anthocyanin biosynthesis during petal color fading, the amounts of metabolites with six anthocyanin components were analyzed: delphinidin, pelargonidin, cyanidin, peonidin, malvidin, and petunidin. A total of 26 cyanidin derivatives were detected. Among these metabolites, 20 out of 26 (76.9%) were decreased from T1/T2 to T3. The remaining metabolites with an increasing trend showed different modifications of chemical groups or sites, such as chloride and pentoside (Figure 5a). Regarding delphinidin, 24 metabolites were detected, and 15 delphinindin derivatives decreased from T1/T2 to T3 (Figure 5b). Regarding pelargonidin, 10 out of 13 (76.9%) decreased from T1/T2 to T3 (Figure 5c). Regarding peonidin, three out of five derivatives showed a decreasing trend (Figure 5d). Regarding malvidin and petunidin derivatives, two out of eight and five out of five showed decreased amounts from T1/T2 to T3 (Figure 5e,f). These results indicate that most anthocyanin derivatives decreased during petal color fading.

### 2.4. Identification and Functional Analysis of Differentially Expressed Genes during Zhehuhong Petal Color Fading

To further reveal the genes involved in the process of petal color fading in Zhehuhong, transcriptomics analysis was performed. The results show that there were 2320 and 2024 distinctive genes at T1 compared to T2 and T2 compared to T3 (Figure 6a). Among the detected genes, 5791 and 6453 genes were upregulated and 6867 and 4680 genes were downregulated from T1 to T2 and T2 to T3, respectively (Figure 6b,c).

The results of further analysis of pathway enrichment show that flavone and flavonol biosynthesis had the highest degree of richness, suggesting the importance of flavonoid metabolism during petal color fading at stages T1 to T2. Other pathway enrichment was mainly related to fatty acid, amino acid, and carbohydrate metabolism (Figure 6d). Carotenoid biosynthesis showed very high pathway enrichment from T2 to T3. Sugar, lipid, and nitrogen metabolism still had very high richness. Among these molecules, many glycolipid and amino acid sugar compounds were found in the activity of metabolism (Figure 6e). In addition to these compounds, phytohormone metabolism was also detected. The above results suggest that the identified genes from T1 to T3 are not only involved in petal cell senescence, but are also related to color fading. In order to further uncover the alterations of genes in the anthocyanin biosynthesis and phytohormone regulation pathways, the transcript levels of selected genes via RNA sequencing were validated by real-time quantitative PCR analysis. The results show that the relationship between them was significant and positively correlated (R^2^ = 0.92) (Figure 6f).

### 2.5. Identification of Genes Involving Anthocyanin Biosynthesis 

The anthocyanin biosynthesis pathway was outlined, and then the expression of genes encoding enzymes at each metabolic node was analyzed via RNA sequencing (Figure 7). The results show that the genes encoding enzymes catalyzing phenylalanine into three types of anthocyanin generally had high relative amounts of transcripts at T1 (Figure 7). This indicates that the color fading of red petals is closely associated with the reduced expression of those key genes from T1 to T3.

### 2.6. Identification of Genes Related to Phytohormone Metabolism during Petal Color Fading

Almost 50% of identified genes encoding the enzymes *S*-adenosylmethionine synthetase (SAMS) and 1-aminocyclopropanecarboxylic acid oxidase (ACO) for ethylene biosynthesis were upregulated from T1/T2 to T3. However, two genes encoding 1-aminocyclopropanecarboxylic acid synthase (ACS) were upregulated from T1 to T3 (Figure 8). For IAA, only one gene, *BnaC05G0025300ZS*, showed increasing expression from T1 to T3. For JA, 13 genes were upregulated while 20 genes were downregulated from T1 to T3. For ABA, 10 out of 17 genes were downregulated from T1 to T3. For GA, four genes were downregulated while three genes were upregulated from T1 to T3. For CTK, two genes, *BnaA09G0543500ZS* and *BnaC08G0389900ZS*, were downregulated from T1 to T3. However, for MT, the reverse trend was found. For SA, all genes encoding two enzymes, ICS1 and PAL1, representing two main SA biosynthesis pathways, were downregulated. However, genes encoding another enzyme, PBS3, were upregulated from T1 to T3 (Figure 8).

### 2.7. Co-Expression Analysis of Metabolites and Genes during Petal Fading

To further understand the relationship between differentially expressed metabolites and genes during fading, co-expression analysis was performed based on a correlation coefficient of more than 0.9 and *p* < 0.05. The results show that there were 244 pairs between cyanidin and its derivatives and corresponding genes. For delphinidin, malvidin, pelagonidin, peonidin, and petunidin and their derivatives, 313, 192, 234, 141, and 92 corresponding genes were identified, respectively (Figure 9a).

Similarly, co-expression analysis was conducted between phytohormone content and corresponding genes based on the correlation coefficient and significance level. The results show that JA, ACC, ABA, GA3, indol-3-acetic acid, indole-3-carboxaldehyde, isopentenyl adenosine, kinetin, melatonin, methyl indole-3-acetate, methyl jasmonate, SA, and trans-zeatin-riboside had 73, 104, 74, 4, 39, 65, 63, 61, 29, 63, 35, 118, and 69 pairs, respectively (Figure 9b).

## 3. Discussion

Flower color is an important trait of rapeseed. During the past decade, the flower color of rapeseed was greatly enriched from yellow to white, orange, red, and purple [26], substantially improving its ornamental value. However, we found that the petal color of Zhehuhong faded markedly from flowering to wilting, whereas the traditional yellow color of petals does not fade. This phenomenon is not new; it also occurs in other ornamental plant species such as *Malus spectabilis* (Ait.) Borkh, *Rosa chinensis Jacq.*, and *Paeonia Suffruticosa* [25,27,28]. However, the mechanism of color fading is largely unknown due to differences in plant species and pigment components. Therefore, revealing the regulating mechanism of petal color fading in rapeseed is meaningful for the breeding of colored rapeseed.

### 3.1. Effect of Morphological Changes of Petal Color on Petal Color Fading

Our results show that the epidermal cells changed from oval to flat and the parenchyma enlarged after merging and then collapsed. The vascular bundles were reduced to a very small size and collapsed as well. Previous investigations reported that the state of epidermal cells is closely related to light absorption and reflection, thus affecting petal coloration [29]. Normally, protruded and full epidermal cells have a larger surface area, which can enhance light absorption by pigment to increase the intensity and brightness of petal color [29,30,31]. However, flat cells reflect more incident light, resulting in a lighter petal color [29]. For example, it was reported that epidermal cells changed from flat to conical, and petal color deepened from pink to purple in *Antirrhinum majus* L. [32,33]. The parenchyma lies between the upper and lower epidermal cells, and cell layers vary among plant species. For example, there is only single cell layer in *Papaver somniferum* L., while there are several cell layers in *Rafflesia arnoldii* [34,35]. It was reported that anthocyanin is also deposited in parenchyma [36]. Therefore, parenchymal collapse might reduce the accumulation of anthocyanin, leading to fading petal color. Vascular bundles are beneficial to maintaining a sufficient water supply in cells [25]. However, as vascular bundles collapse, cell water decreases and anthocyanin becomes diffused in other organelles, which might be an important reason for fading petal color. However, we only observed the structure of petal tissues, and further chemical analyses are required to confirm these hypotheses.

### 3.2. Relationship between Physiological Changes in Petal Color and Color Fading

In our study, we found that petals had weak acidity, with pH decreasing during color fading. A previous investigation revealed that petal color change might not be due to pH change [37]. In *Quisqualis indica* L., the petal color was found to change from white to red, but this was not correlated with pH, as the color was similar to rapeseed in an acidic environment [38]. Therefore, the current results also suggest that color fading might not be caused by a single factor such as pH.

In addition to pH, pigment content is significantly correlated with color fading, especially anthocyanin content [12]. In the current study, the detected pigments—tannins, procyanidins, flavonoids, total phenols, and carotenoids—decreased from T1 to T3, which was in accordance with the petal color fading. Pigment type and content are the main factors affecting petal color [39]. Our previous study showed that cyanidin is the main anthocyanin in Zhehuhong [13]. Although we did not further analyze the pigment composition in detail, the decreased anthocyanin content is powerful evidence proving that color fading is tightly related to decreased anthocyanin content. In a previous investigation, the highest anthocyanin content was found before flowering, with dark pink in *Dendranthema morifolium* (Ramat.) Tzvel. and *Cornus officinalis* Sieb. et Zucc.; however, the color faded to light pink due to the decreased anthocyanin content [40,41]. In Zhehuhong, when plants were edited by *BnaA03.ANS* via RNA interference, the petal color faded to yellow or light red due to the decreased anthocyanin content [12]. This result supports the viewpoint that decreased anthocyanin content can affect petal color whether it is caused by external or internal factors.

In addition to the effect of morphological changes or decreased anthocyanin content, petal pigmentation is also regulated by phytohormones [42,43]. Ethylene is an important phytohormone that is closely associated with flower petal wilting [44]. It was reported that ethylene can inhibit anthocyanin biosynthesis [45]. In this study, we found that the content of 1-aminocyclopropanecarboxylic acid, a precursor of ethylene, significantly increased. This increase should have dual roles in petal wilting and the regulation of anthocyanin content. Salicylic acid is an important phytohormone in plant disease defense [46]. It was revealed that the external application of an appropriate level of SA can double the content of malvidin-3-*O*-*β* glucoside in grape berry [47]. In another report, the contents of delphinidin-3-*O*-glucoside, cyanidin-3-*O*-glucoside, petunidin-3-*O*-glucoside, and malvinidin-3-*O*-glucoside, but not peonidin-3-*O*-glucoside, significantly increased in grape berry with the application of SA at 50 mg L^−1^. However, the study did not show whether the color changed with the application of SA [48].

Although our study found that the SA content increased almost 30 times from T1 to T3, the relationship between SA and petal color fading and their regulation mechanism are still unclear. It is considered that cytokinins modulate anthocyanin accumulation via several pathways, such as sucrose-induced anthocyanin biosynthesis in the presence of light [49,50,51]. In our study, the content of three cytokinins and derivatives significantly decreased during petal fading. This result suggests that decreased cytokinins and anthocyanin have the same roles in petal fading. It is proved that anthocyanin deposition in plant tissue is promoted by jasmonic acid through the interaction of JASMONATE-ZIM-DOMAIN PROTEIN proteins and WD-Repeat/bHLH/MYB complexes including MYB75 in the presence of light [52,53,54]. In this context, the decreased jasmonic acid content during petal fading in the current study was in accordance with the reduced anthocyanin content. Abscisic acid is involved in anthocyanin biosynthesis as well [55]. In apple, MdBT2 protein can degrade an ABA-induced bZIP transcription factor, MdbZIP44, via the ubiquitin-26S proteasome system, thus repressing anthocyanin biosynthesis [56,57,58]. Although we do not know the regulation mechanism of abscisic acid in anthocyanin biosynthesis, the reduced abscisic acid content was synchronous with the anthocyanin content. We suppose that abscisic acid has a role in petal color fading. It was reported that unlike cytokinins, jasmonic acid, and abscisic acid, auxin can repress anthocyanin accumulation [59,60]. However, those results also suggest that the regulation occurs in an IAA-concentration-dependent manner [61]. In our result, as anthocyanin content decreased and petal color faded, the detected auxin content decreased, which seemed to have a positive role in the regulation of petal color fading. In fact, it is recognized that anthocyanin deposition in plant tissue is regulated by many phytohormones, as mentioned above, but the exact regulation network is very poorly understood; thus, the phenomenon of petal color fading is also poorly understood. It is necessary to further determine the function of each phytohormone in the regulation of anthocyanin content and petal color fading in rapeseed.

### 3.3. Joint Multi-Omics Analysis of Petal Color Fading

To further explore the molecular regulation of petal color fading in Zhehuhong, metabolomics analysis was performed. Among the identified differentially expressed metabolites, most of them were related to anthocyanin, flavone, and flavonol biosynthesis. Our previous study showed that cyanidin is a dominant anthocyanin component in Zhehuhong petals and contains small amounts of pelargonidin and peonidin [13]. We further analyzed six anthocyanin components via metabolomics analysis during petal color fading. In general, the contents of cyanidin, pelargonidin, and peonidin and their derivatives showed a decreasing trend, which was in accordance with petal fading and in agreement with the measured anthocyanidin from T1 to T3. In our previous study, we did not detect malvidin or petunidin in Zhehuhong petals [13]; however, there were substantially fewer derivatives of those compounds than for cyanidin, delphinidin, and pelargonidin (except peonidin) in this study. Furthermore, only one component of both anthocyanins showed an increasing trend, while all others showed a decreasing trend; therefore, if both exist in Zhehuhong petals, it can be inferred that they are involved in petal fading. As for delphinidin, the result was different from our previous study: 24 delphinidin derivatives were identified in the present study [5]. The most likely reason is that only one compound, delphinidin chloride, was detected, but it was not a comprehensive analysis. Therefore, the fact that we did not detect delphinidin chloride does not mean that other derivatives do not exist. Interestingly, 70.8% of derivatives showed a decreasing trend. Thus, the results also support the viewpoint that delphinidin derivatives have an important role in petal fading. Decreased pigment content, especially anthocyanin content, is an important factor in plant tissue fading, as mentioned above. In *Nelumbo nucifera*, the anthocyanin content was found to be reduced from 62 to 3 µg g^−1^ during the fading of petals, which turned from red to almost white. Furthermore, delphinidin 3-*O*-glucoside, cyanidin 3-*O*-glucoside, petunidin 3-*O*-glucoside, and peonidin 3-*O*-glucoside also showed a decreasing trend [24]. Similar investigations were also reported [62,63]. Therefore, our metabolomics analysis once again proved the importance of anthocyanin content in petal coloration.

In addition to metabolomics, transcriptomics analysis was also performed. Our results show that the genes involved in flavone and flavonol biosynthesis had the highest richness; however, those genes were downregulated from T1 to T2. This result indicates that substantial downregulation of gene expression will cause a reduction in anthocyanin content. The gene expression in the whole process of anthocyanin biosynthesis showed that almost all identified genes in this study, especially *DFR*, *ANS*, *UFGT*, and *UF3GT*, had higher expression at T1 and then significantly decreased at T2 and T3. This result directly supports the viewpoint that anthocyanin biosynthesis is closely associated with the reduction in corresponding genes. To date, many genes associated with anthocyanin biosynthesis have been isolated and their function in the regulation of anthocyanin biosynthesis has been elucidated [64]. However, the modulation of anthocyanin degradation has not been fully investigated. In pear, *PbLAC4-like* gene functions to regulate anthocyanin degradation through activation of *PbMYB26* during fading [65]. A study of *Malus hupehensis* petals fading from red to white found significantly higher expression levels of *PAL*, *CHS*, *CHI*, *DFR*, *FLS*, *ANS*, *UFGT*, *MYB10*, and *MYB12* at early development stages than later stages. The authors further inferred that *MBY10* regulates anthocyanin biosynthesis by affecting the expression of *ANS* [66]. Therefore, the regulation of color fading will most likely be determined by focusing on MYB transcriptions activating or deactivating anthocyanin biosynthesis genes. Furthermore, the limited genes that have been identified to date suggest that many studies on gene isolation and function in the color fading mechanism will be required in the future.

As for the genes encoding phytohormone biosynthesis, many key genes were identified during color fading in Zhehuhong. As an example, we took a gene from jasmonic acid biosynthesis. Jasmonic acid enhancing anthocyanin biosynthesis has not been under debate during the past decades. Four genes encoding JASMONATE-ZIM-DOMAIN PROTEIN (JAZ) were identified in the current study. Except for one gene, *Bna05G0015600ZS*, which showed a slight increase from T2 to T3, the other three genes exhibited a decreasing trend. Preliminary reduction in JAZ expression may be an important regulatory pathway leading to petal fading according to a previous study [67], but further evidence should be provided by a series of gene function experiments.

Because of the comprehensive regulation of genes to metabolites, we conducted joint metabolomics and transcriptomics analysis. Our results show that the metabolites mainly associated with anthocyanin have many significantly correlated genes. For example, 41 genes are positively and significantly associated with cyanidin. After assaying those genes further, we found that the genes encoding enzymes are distributed from phenylalanine to cyanidin biosynthesis. Some genes had several alleles to finish the process of fading. Although a previous study showed that a non-corresponding relationship was always found between metabolomics and transcriptomics [68], a considerable pairwise correlation between metabolites and genes was found in this study. One important reason might be the relatively simple metabolic activity during petal fading. The good result of the current study might provide an analytic model for joint multi-omics analysis.

This study had two goals: to explore the mechanism of petal color fading and provide possible breeding strategies for color-stable rapeseed varieties. Regarding the former, a number of morphological, physiological, and gene expression level alterations were observed; however, the linkage between the changes in these indices at different levels and fading need further experiments for validation. Regarding the latter, one breeding strategy would be to create germplasm with other pigment components that are less sensitive or not sensitive to environmental factors such as temperature and light intensity. Another way would be to manipulate genes, such as blocking ethylene production to slow down petal wilting and simultaneously enhancing the expression of genes encoding jasmonic acid production to activate anthocyanin biosynthesis.

## 4. Materials and Methods

### 4.1. Plant Materials

The plant materials in this study were from the Zhehuhong red flower rapeseed line, provided by the Zhejiang Academy of Agricultural Sciences. The experiment was carried out at the experimental station of the Zhejiang Academy of Agricultural Sciences on 16 October 2021 and 21 October 2022. The planting method was direct seeding with about 5 seeds in each hole. When the plants grew to the 5-leaf stage, the seedlings were removed, and one plant was left in each hole. A slow-release compound fertilizer (N: P_2_O_5_: K_2_O = 15:7:8) with 15 kg ha^−1^ boron (B) was applied to the soil as the base fertilizer in the amount of 750 kg ha^−1^. The fertilizer was purchased from Hubei Yishizhuang Agricultural Science and Technology Co., Ltd. (Yichang, Hubei, China). The plants were not irrigated with water during the growth period. Insect pest management was consistent with local management practices.

### 4.2. Experimental Design

The trial was a fully randomized block design with 3 replicates. The experiment comprised 3 treatments: 2 days before flowering (T1) for petal sampling, and at this stage flowers were in bud status; at flowering (T2) for petal sampling, and at this stage flowers were just blooming; and 2 days after flowering (T3), and at this stage the petals appeared to be withered. The plot area was 18 m^2^ with 15 cm plant space and 30 cm row space.

### 4.3. Sampling

Plants were sampled using a destructive method at stages T1, T2, and T3. About 20 flowers were randomly selected per plot during sampling, and the sampling process excluded border plants to avoid a marginal effect. The flowers were sampled at the flowering stage (including 2 days before flowering) between the end of February and early March each year, when the temperature was relatively low, to minimize the effects of temperature on petal pigment metabolism. The flowers were only selected from the main inflorescence at a sunny time in the morning. During sampling, buds (T1) or flowers (T2 and T3) were picked, and the bracts were immediately removed by sterilized and RNase-free tweezers. The tweezers were used to gently pick the petals at the base, and one piece was immediately frozen in liquid nitrogen for phytohormone content determination and metabolomics/transcriptomics analysis. A second piece of the sample was immediately put into Canoy’s fixative solution for paraffin sectioning and glutaraldehyde buffer fixative solution for ultrastructure analysis by scanning electron microscopy. A third piece of the sample was immediately brought back to the laboratory for the measurement of pH and tannin, procanthocyanin, flavonoid, carotenoid, total phenol, and anthocyanin content.

### 4.4. Observation of Petal Morphology by Stereomicroscope, Paraffin Section, and Scanning Electron Microscopy

The morphology of fresh petals from the 3 stages (T1 to T3) was photographed and recorded using a 10× stereomicroscope (VHX-950F, Keyence, Osaka, Japan). Next, the petals fixed by Carnoy’s fixative solution were paraffin sectioned. The main process was as follows: a fresh petal was immediately put into Canoy’s fixative solution (anhydrogenic ethanol: glacial acetic acid = 3:1 volume), and then stored at room temperature. The tissue was removed from the fixative solution in the embedded box using tweezers. The dehydration box was placed in the dehydration machine with gradient alcohol for dehydration and immersed in melted paraffin. Then, the wax-soaked tissue was embedded in the embedding machine, and the wax block was trimmed. The trimmed wax blocks were placed in a paraffin microtome and sliced 4 μm thick, and the sections were successively dewaxed with water and then scanned with neutral gum seal.

The ultrastructure of petals during the fading process was recorded by scanning electron microscopy (Hitachi Regulus 8100, Tokyo, Japan). The main process was as follows: petals fixed with glutaraldehyde phosphate buffer (pH 7.2, 0.2 mol L^−1^) for 24 h were cut into small squares and rinsed 3 times with phosphate buffer (pH 7.2, 0.1 mol L^−1^) for 20 min each time. Then, the petals were dehydrated with alcohol gradient (30%, 50%, 70%, and 90%), and treated once per concentration for 15 min. Finally, the petals were repeatedly dehydrated with 100% alcohol 3 times, for 15 min each time. After that, CO_2_ critical point drying was carried out, and a gold-spraying instrument was used for electron microscope scanning observation.

### 4.5. Determination of pH Value and Pigment Content in Petals

To determine the pH value of petals, about 0.5 g of the sample was quickly weighed and ground into powder in liquid nitrogen. Then, the powder was transferred into a centrifuge tube with deionized water up to 10 mL. Each sample was centrifuged at 12,000 rpm min^−1^ for 15 min. Then, the supernatant was measured using a pH meter (FE28, Mettler-Toldeo, Zurich, Switzerland). The contents of tannins, proanthocyanidins, flavonoids, carotenoids, total phenols, and anthocyanins in petals at the 3 stages were measured using a testing kit (Suzhou Komin Biotechnology Co., Ltd., Suzhou, China) according to the instructions.

### 4.6. Determination of Phytohormone Content in Petals

Samples of about 25 mg were weighed and put into 2 mL centrifuge tubes containing a steel ball and 1 mL of isopropyl alcohol aqueous solution (80:20, *v*/*v*, internal standard mixture containing isotope labels). The mixture was swirled for 30 s and homogenized for 4 min at 40 Hz. Then, the homogenate was treated by ultrasound 3 times for 5 min in an ice water bath. The homogenate was left to stand at −20 °C for 1 h and centrifuged at 14,000 rpm min^−1^ for 15 min. Then, 800 μL of supernatant was taken, rotated, and concentrated until dry, and the extracts were re-dissolved with 160 μL of methanol aqueous solution (50:50, *v*/*v*).

Finally, the solute was centrifuged at 14,000 rpm min^−1^ for 10 min, and the extract was obtained after passing through a 0.22 μm filter membrane. The target compounds were isolated using an ACQUITY HSS T3 UPLC liquid chromatography column (100 × 2.1 mm, 1.8 μm; Waters) ACQUITY I-Class ultra-high performance liquid chromatography system (Waters Technologies Shanghai Co., Ltd., Shanghai, China). Mass spectrometry was performed in multiple reaction monitoring (MRM) mode using a SCIEX QTRAP 6500 with IonDrive Turbo V ESI ion source and a triple quadrupole mass spectrometer (SCIEX, Framingham, MA, USA).

### 4.7. Metabolomics Analysis

Samples of about 50 mg were weighed and 1000 μL of extraction solution and magnetic beads were added to the 2 mL tubes for grinding. The homogenate was treated by ultrasound, and the supernatant was taken after static centrifugation to obtain the metabolite extraction. The analysis was performed using an ACQUITY I-Class PLUS ultra-HPLC with Xevo G2-XS QTOF (Waters, Milford, MA, USA) high-resolution mass spectrometer. The chromatography was performed on an ACQUITY HSS T3 HPCL column (1.8 μm 2.1 × 100 mm). The original data, including peak extraction and peak alignment, collected by a MassLynx V4.2 were processed by Progenesis QI software (v2.4.69111.27652). Theoretical fragment identification of metabolites was carried out using the online Progenesis QI METLIN database, a public database, and a self-built Tanyuan database. The mass number deviation of the parent ion was 100 mg kg^−1^, and the mass number deviation of the fragment ion was less than 50 mg kg^−1^. A fold change ≥ 1 and *p* < 0.05 indicated differentially accumulated metabolites (DAMs). Venn and volcano diagrams, principal component analysis (PCA), Gene Ontology (GO), and Kyoto Encyclopedia of Genes and Genomes (KEGG) were used to further analyze the metabolomic changes of pigment in rapeseed petals at different development stages.

### 4.8. RNA-Seq Analysis

RNA was extracted from samples, and RNA purity and concentration were detected using a NanoDrop 2000 spectrophotometer (ThermoFisher Scientific, Waltham, MA, USA) and RNA integrity was detected using an Agilent 2100/LabChip GX (Agilent, Santa Clara, CA, USA). The insert piece of the library was tested using the Qsep400 high-throughput analysis system. PE150 mode sequencing was performed on the Illumina NovaSeq6000 sequencing platform (Illumina, San Diego, CA, USA). The offline data were filtered to obtain clean data, and the sequence was compared with the specified reference genome to obtain mapped data. A fold change ≥ 2 and FDR < 0.01 indicated differentially expressed genes (DEGs). Venn and volcano diagrams, PCA, GO, and KEGG were used to analyze the transcriptomic changes of pigment in rape petals at different development stages.

### 4.9. Quantitative Real-Time Polymerase Chain Reaction (qRT-PCR) Analysis

To verify the reliability of the RNA-Seq data, 16 DEGs were selected for qRT-PCR testing. The sequences of primers are listed in Appendix A. The RNA of petals from the 3 development stages was extracted using an RNeasy Plant Mini Kit (Qiagen, Dusseldorf, Germany), and the extracted total RNA was reverse-transcribed to synthesize cDNA using a PrimeScript^TM^ 1st Strand cDNA Synthesis Kit (Takara Bio, Kyoto, Japan). The chosen genes were *PAL*, *C4H*, *4CL*, *CHI*, *F3H*, *F3′ H*, *FLS*, *DFR*, *ANS*, *UFGT*, *UGT75C1*, and *UF3GT*. *BnACTIN* was used as the reference gene.

### 4.10. Statistics

SPSS statistical software (version 24.0, Chicago, IL, USA) was used for analysis of variance (ANOVA). The contents of tannins, proanthocyanidins, flavonoids, carotenoids, total phenols, anthocyanins, and hormones in petals were expressed as mean values, and significant differences among treatments were analyzed using Duncan’s method (*p* < 0.05). The transcript levels were calculated by 2^−ΔΔCt^ (Livak et al., 2001 [69]).

## 5. Conclusions

The current study revealed that the morphology of petal cells in Zhehuhong, including the epidermis, parenchyma, and vascular bundles, significantly collapses during petal wilting, implying that pigments such as anthocyanins released from those tissues result in color fading. The cellular pH value and all detected contents of pigments significantly decreased, suggesting that anthocyanin degradation plays a vital role in petal color fading. Phytohormones, including 1-aminocyclopropanecarboxylic acid and salicylic acid, significantly increased, while most of the others decreased. The more than 90% decrease in jasmonic acid and abscisic acid content validated their positive regulation of anthocyanin degradation. Both metabolomic and transcriptomic analysis indicated that the early downregulation of metabolites and genes in anthocyanin biosynthesis was direct evidence of petal color fading. Furthermore, the results also show that it is essential to choose the appropriate stage for genetic manipulation to block petal color. The co-expression analysis of metabolites and genes showed that certain metabolites had a number of significantly associated genes, which were distributed in the metabolic pathway and could be further screened as candidate genes for functional study. As an example, two alleles encoding dihydroflavonol 4-reductase, *BnaA09G0187400ZS* and *BnaC09G0215200ZS*, had correlation coefficients of 0.953 and 0.952, respectively, relative to the content of cyanidin. The two candidate genes could be further focused on their regulation on cyanidin degradation. The co-expressed analysis for multiple-omics extended a useful candidate gene screening system. Our results provide new insights on further elucidating the function of the key genes regulating petal color fading and genetic manipulation and breeding of stable petal colored rapeseed varieties.

## Figures and Tables

**Figure 1 ijms-25-02577-f001:**
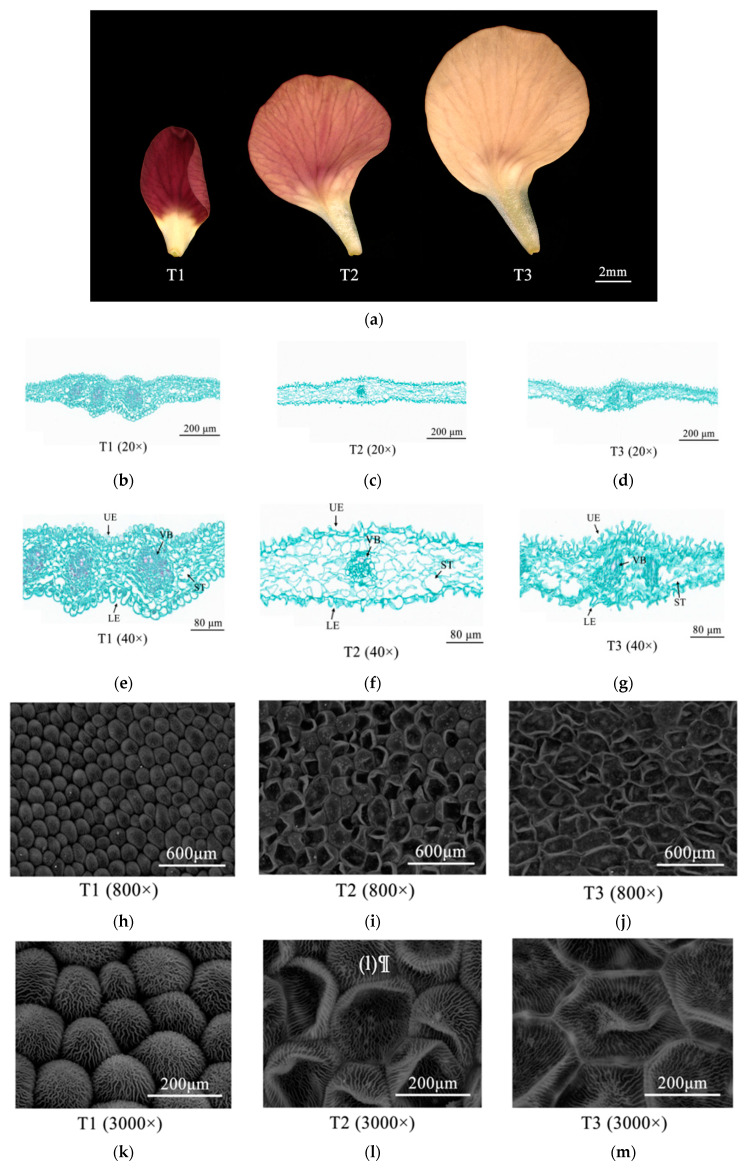
Morphological changes of Zhehuhong petal cells 2 d before flowering (T1), at flowering (T2), and 2 d after flowering: (**a**) under stereomicroscopy; (**b**–**d**) by paraffin section at 20×, from T1 to T3; (**e**–**g**) by paraffin section at 40×, from T1 to T3; (**h**–**j**) by scanning electron microscope at 800×, from T1 to T3; (**k**–**m**) by scanning electronic microscopy at 3000×, from T1 to T3. LE, lower epidermal cell; ST, sponge tissue; UE, upper epidermal cell; VB, vascular bundle.

**Figure 2 ijms-25-02577-f002:**
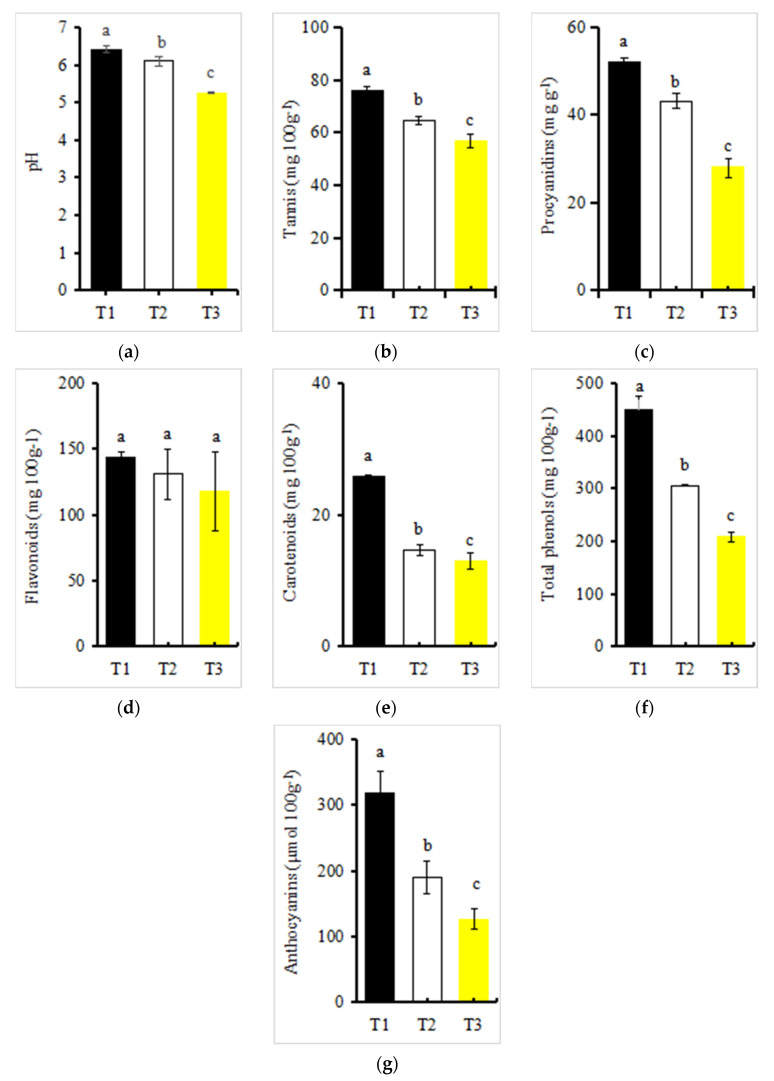
pH value and pigment content changes during petal color fading in Zhehuhong (T1 to T3): (**a**–**g**) pH, tannins, proanthocyanidins, flavonoid, carotenoid, total phenols, and anthocyanin, respectively. Different lowercase letters indicate significant differences between treatments using Duncan’s method (*p* < 0.05). Error bars indicate SD. T1, T2, and T3: 2 d before flowering, at flowering, and 2 d after flowering.

**Figure 3 ijms-25-02577-f003:**
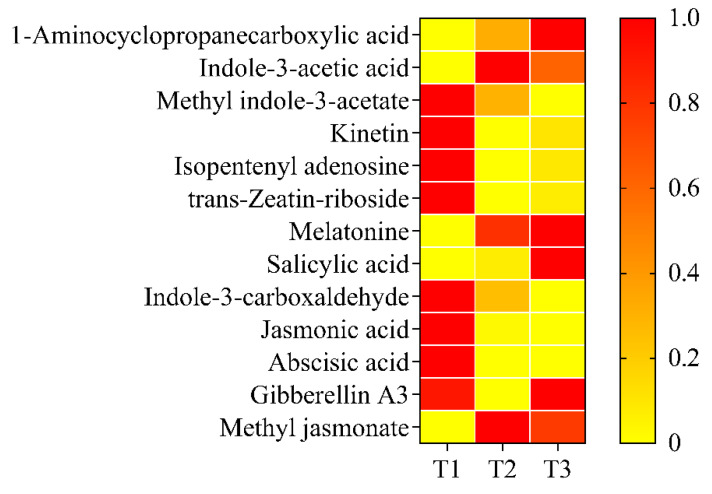
Heatmap of content of 13 phytohormones from T1 to T3. Detected phytohormones included abscisic acid, 1-aminocyclopropanecarboxylic acid, gibberellin A3, indole-3-acetic acid, indole-3-carboxaldehyde, isopentenyl adenosine, jasmonic acid, kinetin, melatonin, methyl indole-3-acetic, methyl jasmonate, salicylic acid, and trans-zeatin-riboside. T1, T2, and T3: 2 d before flowering, flowering, and 2 d after flowering. Darker red indicates higher content while darker yellow indicates lower content.

**Figure 4 ijms-25-02577-f004:**
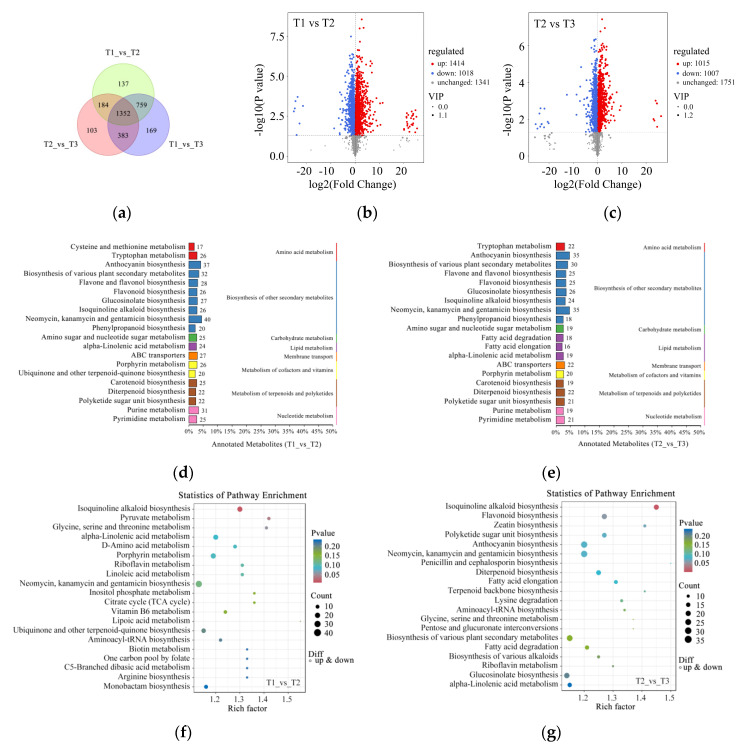
Metabolomics analysis of petal color fading from T1 to T3: (**a**) Venn diagram of metabolites at T1, T2, and T3. (**b**,**c**) Volcano plots indicating metabolite numbers at T1 vs. T2 and T2 vs. T3. Red dots indicate upregulated metabolites, blue dots indicate downregulated metabolites, and gray dots indicate unchanged metabolites. (**d**,**e**) Classification of significantly differentially expressed metabolites at T1 vs. T2 and T2 vs. T3 by KEGG analysis. (**f**,**g**) Rich factor analysis of significantly differentially expressed metabolites at T1 vs. T2 and T2 vs. T3. Dots indicate significantly differentially expressed metabolites, size of dots indicates metabolite counts, and color of dots indicates *p*-value. T1, T2, and T3: 2 d before flowering, flowering, and 2 d after flowering.

**Figure 5 ijms-25-02577-f005:**
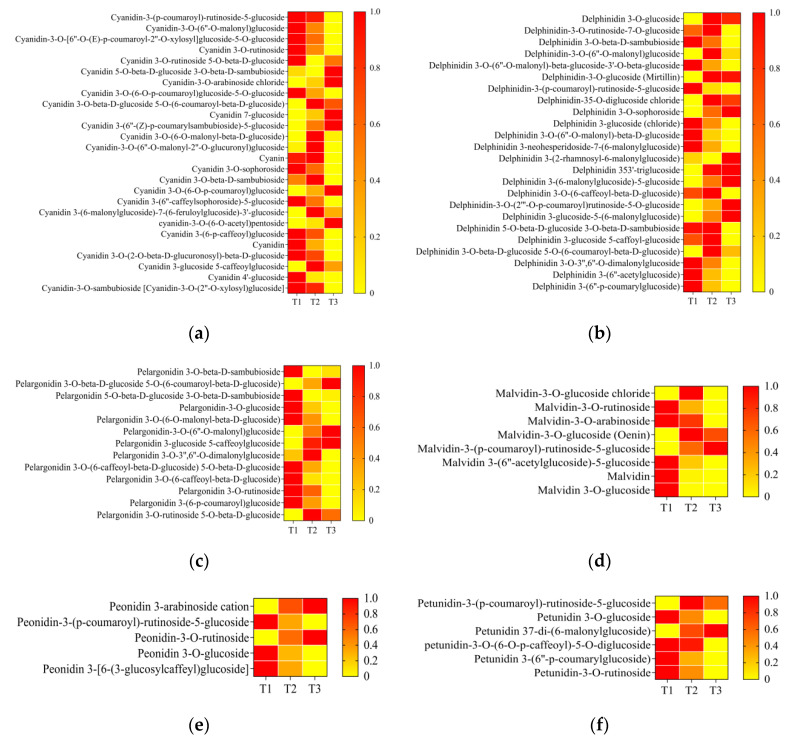
Heatmaps of content of six anthocyanins ((**a**) cyanindin, (**b**) delphinidin, (**c**) pelargonidin, (**d**) malvidin, (**e**) peonidin, and (**f**) petunidin) and their derivatives from T1 to T3. T1, T2, and T3: 2 d before flowering, flowering, and 2 d after flowering. aDarker red indicates higher content; darker yellow indicates lower content.

**Figure 6 ijms-25-02577-f006:**
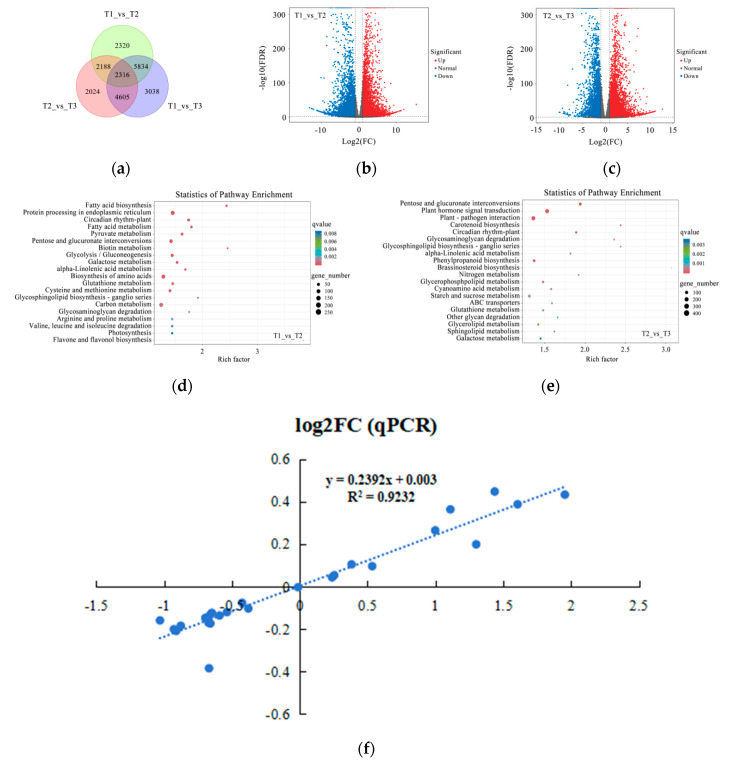
Transcriptomics analysis of petal color fading from T1 to T3: (**a**) Venn diagram of genes detected at T1, T2, and T3. (**b**,**c**) Volcano plots indicating gene numbers at T1 vs. T2 and T2 vs. T3. Red dots indicate upregulated genes, blue dots indicate downregulated genes, and gray dots indicate unchanged genes. (**d**,**e**) Rich factor analysis of significantly differentially expressed genes at T1 vs. T2 and T2 vs. T3. Dots indicate significantly differentially expressed metabolites, size of dots indicates metabolite counts, and color of dots indicates *p*-value. (**f**) Correlation analysis of selected genes between results of quantitative polymerase chain reaction (qPCR) and transcriptomics analysis. T1, T2, and T3: 2 d before flowering, flowering, and 2 d after flowering.

**Figure 7 ijms-25-02577-f007:**
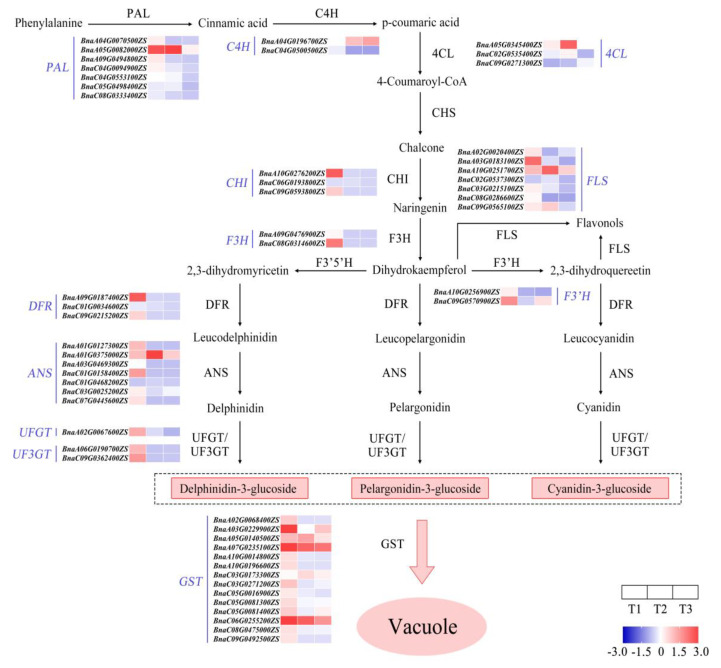
Outline of gene expression analysis of anthocyanin biosynthesis during petal color fading from T1 to T3. T1, T2, and T3: 2 d before flowering, flowering, and 2 d after flowering. Darker red indicates higher content; darker blue indicates lower content. ANS, anthocyanin synthase; C4H, cinnamate 4 hydroxylase; CHI, chalcone isomerase; CHS, chalcone synthase; 4CL, 4-coumaryl:CoA ligase; DFR, dihydroflavonol 4-reductase; F3H, flavanone 3-hydroxylase; F3′H, flavanone-3′-hydroxylase; F3′5′H, flavanone 3′5′-hydroxylase; FLS, flavonol synthase; GST, glutathione S-transferase; PAL, phenylalanine ammonia lyase; UFGT, flavonoid3-O-glucosyltransferase.

**Figure 8 ijms-25-02577-f008:**
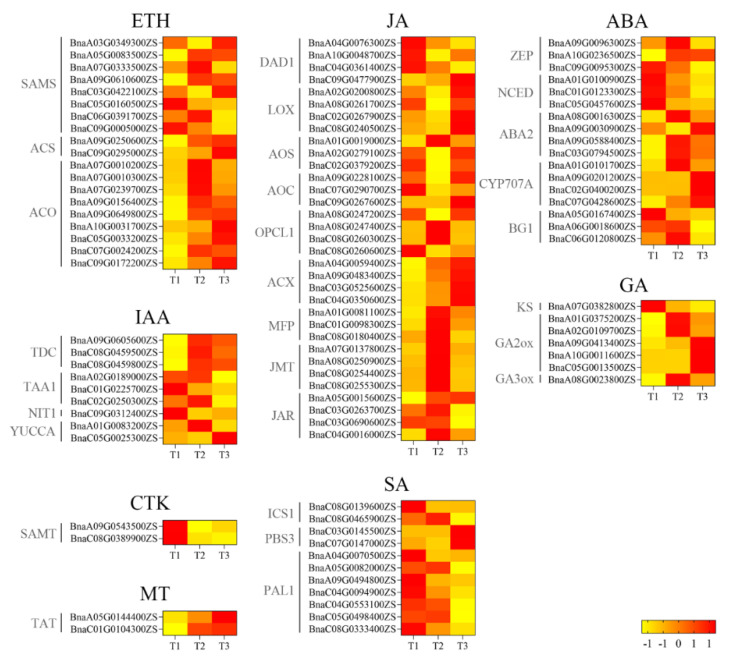
Heatmaps of expression levels of genes encoding phytohormones from T1 to T3. T1, T2, and T3: 2 d before flowering, flowering, and 2 d after flowering. Darker red indicates higher content; darker yellow indicates lower content. ABA, abscisic acid; CTK, cytokinin; ETH, ethylene; GA, gibberellin; IAA, indole-3-acetic acid; JA, jasmonic acid; MT, melatonin; SA, salicylic acid.

**Figure 9 ijms-25-02577-f009:**
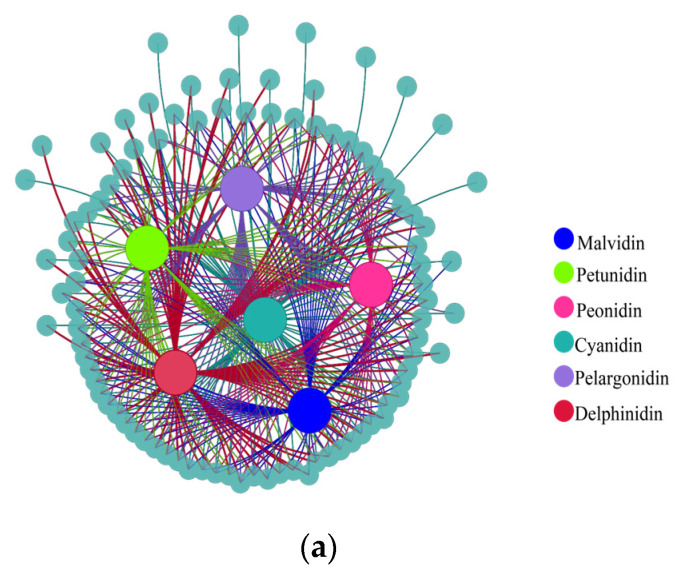
Correlation analysis of (**a**) metabolites and their corresponding genes and (**b**) phytohormones and their corresponding genes. Large solid colored circles indicate different metabolites or phytohormones, small solid gray-blue circles indicate their corresponding genes. Lines between large and small solid circles indicate positive correlations. Data were selected based on *p* < 0.05 and correlation coefficient more than 0.9.

## Data Availability

Data are available by reasonable request from the corresponding author.

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
