# Peer review of "Multi-Omics Analysis Reveals That Anthocyanin Degradation and Phytohormone Changes Regulate Red Color Fading in Rapeseed (Brassica napus L.) Petals"

_ijms, 2024, doi:10.3390/ijms25052577_

Round 1

Reviewer 1 Report

Comments and Suggestions for Authors

This manuscript by Huang et al. investigates the morphological, physiological, metabolomic, and transcriptomic changes underlying petal colour fading in the red-flowered rapeseed variety Zhehuhong. Using a combined multi-omics approach, the study aims to shed light on the regulation of anthocyanin degradation and petal senescence in coloured rapeseed. The authors find that decreased anthocyanin content resulting from reduced expression of biosynthetic genes is a key driver of petal fading. Alterations in phytohormones and petal cell structure also appear to play contributing roles in the process. Overall, the results enhance our mechanistic understanding of petal colour instability during flower development in Brassica napus

However, there are several concerns which need to be addressed for better understanding of the readers. 

·      The study only examined one coloured rapeseed variety (Zhehuhong). Analyzing multiple genotypes with different petal colours would allow better generalization of the results. (Lines 339-341)

·      Anthocyanin levels were quantified as total anthocyanin content. Profiling individual anthocyanin compounds could provide more insights into the specific pigments degraded during fading. (Lines 356-358)

·      Phytohormone and transcriptomic analyses were conducted at three snapshot time points. Additional sampling over the full-time course of fading would give a higher resolution on the dynamics of changes. 

·      Functional validation of key genes was not performed. Knock-out/overexpression studies could help validate proposed roles in regulating petal fading. 

·      Environmental factors like temperature and light that can influence anthocyanin stability were not evaluated or controlled in this study. or if there were any particulars about the experiment, provide the details in the methodology section.

·      The relationship between anthocyanin degradation and petal cell structural changes was hypothesized but not demonstrated. Additional cytochemical experiments could show the localization of pigment loss. (Lines 344-346)

·      Mechanistic links between altered phytohormone levels and effects on anthocyanin metabolism require further elucidation. (Lines 419-421)

·      Metabolite-transcript correlations provide hints but do not prove regulatory relationships. More in-depth network/pathway analysis could be insightful. (Lines 489-491)

·      The study is descriptive and does not provide strategies for breeding color-stable varieties. Suggestions for applications could be discussed. 

·      The role of anthocyanin transport and vacuolar sequestration in petal fading was not addressed. (Lines 359-360)

Author Response

This manuscript by Huang et al. investigates the morphological, physiological, metabolomic, and transcriptomic changes underlying petal colour fading in the red-flowered rapeseed variety Zhehuhong. Using a combined multi-omics approach, the study aims to shed light on the regulation of anthocyanin degradation and petal senescence in coloured rapeseed. The authors find that decreased anthocyanin content resulting from reduced expression of biosynthetic genes is a key driver of petal fading. Alterations in phytohormones and petal cell structure also appear to play contributing roles in the process. Overall, the results enhance our mechanistic understanding of petal colour instability during flower development in Brassica napus. 

R: Thank you for your positive comments.

However, there are several concerns which need to be addressed for better understanding of the readers. 

  • The study only examined one coloured rapeseed variety (Zhehuhong). Analyzing multiple genotypes with different petal colours would allow better generalization of the results. (Lines 339-341)

R: Sure. I completely agree with you. However, till now, we found that the yellow rapeseed is very stable. Sometimes, white rapeseed is not stable but had been clearly studied by researchers that were cited in the manuscript. Although there were orange and purple rapeseed, they are much stable than red one. We only have one red rapeseed line that has very significant fading phenomenon. Therefore, no multiple genotypes can be selected at current stage. I hope you can understand this situation. Thank you.

  • Anthocyanin levels were quantified as total anthocyanin content. Profiling individual anthocyanin compounds could provide more insights into the specific pigments degraded during fading. (Lines 356-358)

R: Very good suggestion and it would provide a very clear result. However, we found that there were many derivatives for the six kinds of anthocyanin compounds, therefore, we used metabolomic data to dig data of individual anthocyanin compounds, which should be effective as well. Anyway, this is a very good advice.

  • Phytohormone and transcriptomic analyses were conducted at three snapshot time points. Additional sampling over the full-time course of fading would give a higher resolution on the dynamics of changes. 

R: Thank you for your suggestion. We observed the process of color fading was in an ever-lasting status from dark red to light red, therefore, we selected three points for phytohormone contents determination and transcriptomic analysis. The three stages were selected at 2 days before flowering, 0 day flowering and 2 days after flowering not only represented the petal color fading but also was in accordance with the duration for petal senescence. The full-time course of monitoring would provide a higher resolution on the dynamics of changes, but I think this requires some specific instruments and cost-consuming. I hope we can get your understanding.

  • Functional validation of key genes was not performed. Knock-out/overexpression studies could help validate proposed roles in regulating petal fading. 

R: Good suggestion. In current study, we only focused on its morphological, physiological, metabolomic and transcriptomic analysis for petal color fading. Although we have screened some interested targeted genes, further functional validation of those genes will be conducted by over-expression and knock out studies to elucidate the regulatory mechanism.

  • Environmental factors like temperature and light that can influence anthocyanin stability were not evaluated or controlled in this study. or if there were any particulars about the experiment, provide the details in the methodology section.

R: This is a good question that we should consider. Our study was an open-field experiment, which is difficult to control environmental factors. However, The flowers were sampled at the flowering stage (including 2 days before flowering), between end of February and early March each year, the temperature relative low at this stage to minimize the effects of temperature on petal pigment metabolism. And the flowers were only selected from the main inflorescence at sunny day in the morning We add this detail in the methods section. For the effect of environmental factors on anthocyanin stability will be further studied under controlled condition such as plant growth chamber using in vivo and in vitro methods.

  • The relationship between anthocyanin degradation and petal cell structural changes was hypothesized but not demonstrated. Additional cytochemical experiments could show the localization of pigment loss. (Lines 344-346)

R: Thank you for your question. In present study, we only observed the petal cell structure changes during fading. And then we use previous literature to ensure whether they have relationships. According to the previous results, they might have close relationship between cell structure changes and pigment deposition, therefore, we hypothesized they also have in rapeseed. However, we did not any experiments to prove this viewpoint in the duration this study performance. I think, in future investigation on this issue would be interest and provide strong evidences to support there linkage.

  • Mechanistic links between altered phytohormone levels and effects on anthocyanin metabolism require further elucidation. (Lines 419-421)

R: Thank you for your suggestion. In current study, we detected almost all kinds of phytohormones. The contents of those phytohormones normally changed consistent with previous study. Those elucidations were placed in the discussion section. However, the mechanistic links especially for experimental evidences were not provided in this study because we did not do any further functional study. In fact, the information on the association between phytohormone content and anthocyanin metabolism are not much. The previous investigations showed that the least debate was the effect of jasmonic acid on the enhancing of anthocyanin. Therefore, we often use jasmonic acid as an example to elucidate their possible relationships. However, it is not easy to elucidate the function of one gene clearly in rapeseed because many alleles in rapeseed for one gene will bring it much more complex. And I think, further investigations on the function of key genes for some selected phytohormones such as ethylene, jasnomic acid, and abscisic acid will be done.

  • Metabolite-transcript correlations provide hints but do not prove regulatory relationships. More in-depth network/pathway analysis could be insightful. (Lines 489-491)

R: Yes, you are right. Metabolite-transcript correlations do not prove regulatory relationships. But it is a useful tool to confirm the genes significantly related with metabolite and further screening of key genes in its biosynthetic pathway. The screened genes can be designed for further functional study to ensure their regulations on the petal fading. In fact, we provided pathway analysis in Figure 7 to attempt to explore the gene regulations on anthocyanin biosynthesis.

  • The study is descriptive and does not provide strategies for breeding color-stable varieties. Suggestions for applications could be discussed. 

 R: Thank you for your suggestion. We add some information on the strategies for breeding color-stable varieties in the discussion section. The goals of the study are dual including exploring the mechanism of petal color fading and providing possible breeding strategies for color-stable rapeseed varieties. For the former, a number of alterations from morphological, physiological, and gene express level were observed, however, the linkage underline between changes of those indices at different levels and fading need a lot of experiments to further validation. For the latter, one way is to create germplasm with other pigment component that less or no sensitive to environmental factors such as temperature and intensity of light. The second way is to manipulate genes such as blocking ethylene production to slow down petal wilting and simultaneously enhancing expression of genes encoding jasnomic acid production to activate anthocyanin biosynthesis.

  • The role of anthocyanin transport and vacuolar sequestration in petal fading was not addressed. (Lines 359-360)

R: Yes, anthocyanin transport and vaculoar sequestration in this study did not been involved. Because such study need the technology of transmission electron microscope and histochemistry staining, we did not consider during design this experiment. However, it can be further investigate in future study. We thank the reviewer’s good suggestion.

Reviewer 2 Report

Comments and Suggestions for Authors

The paper submitted by Huan et al. shows an important advancement in the field of anthocyanins focusing on the pigments in rapeseed flowers. Although widely known by their use in the edible oil industry, there is an increasing interest in rapeseed flowers for ornamental applications. This paper evaluated changes in anthocyanin content, physiological characteristics, and physicochemical parameters of rapeseed flower petals. This was combined with metabolomic and transcriptomic analyses that further expand our understanding on anthocyanin formation and degradation in flowers.   

The paper does a good job in describing the results and discuss them properly. However, rewording is necessary to highlight the importance and practical implications of this study. I believe that one of the most important findings of this study is that a correlation between metabolomics and transcriptomics was found. I believe this is an important finding in this study and should be better highlighted in the conclusions section.

The manuscript's English quality is not adequate for publication. Thorough revision and corrections should be conducted before considering this paper for publication at IJMS. There are several typos and grammar mistakes in the manuscript body and I am certain there are several ones I missed.

Some specific comments:

L050 – L052 could be reworded. Grammar seems off.

L055 – L056 could be reworded.

L083 Anthocyanins are the main pigments…

L084 please clarify what anthocyanin characteristics.

L085 “kind of unstable” is not accurate, are they less stable than other pigments? Please rephrase.

L093 “light flowers”. I believe it should be lightly colored.

L099 “is not only occurred” should be “does not only occur…”

L109 there is no clear description of the objective of this study. This paragraph is a recollection of the methods used and not a concise experimental objective. Please reword.

L157 needs rewording to improve conciseness.

L160 – L162 needs better wording to improve clarity. Result presentation is confusing and a rewording is required. A better reference to Figure 2 in the text can aid in better describing the results.

L442 “…chloride do not mean…” should be “…does not mean…”

L557 Title has a typo. Method is written in present tense, it should be written in past tense.

L573 Method is also written in present tense.

L615 The conclusion is a recollection of results; however, it does not highlight the importance of this work and the repercussions of the information conveyed in the paper. Rewording is needed in the conclusion section to improve the overall quality of the paper.

Comments on the Quality of English Language

Overall, the manuscript's English quality is not adequate for publication. Thorough revision and corrections should be conducted before considering this paper for publication at IJMS. There are several typos and grammar mistakes in the manuscript body and I am certain there are several ones I missed.

Author Response

Responses to Reviewer 2

The paper submitted by Huan et al. shows an important advancement in the field of anthocyanins focusing on the pigments in rapeseed flowers. Although widely known by their use in the edible oil industry, there is an increasing interest in rapeseed flowers for ornamental applications. This paper evaluated changes in anthocyanin content, physiological characteristics, and physicochemical parameters of rapeseed flower petals. This was combined with metabolomic and transcriptomic analyses that further expand our understanding on anthocyanin formation and degradation in flowers.   

The paper does a good job in describing the results and discuss them properly. However, rewording is necessary to highlight the importance and practical implications of this study. I believe that one of the most important findings of this study is that a correlation between metabolomics and transcriptomics was found. I believe this is an important finding in this study and should be better highlighted in the conclusions section.

R: very good suggestion. We have revised the manuscript to highlight the importance and practical implications of the study especially for the correlation between metabolomics and transcriptomics in the conclusions section.

The manuscript's English quality is not adequate for publication. Thorough revision and corrections should be conducted before considering this paper for publication at IJMS. There are several typos and grammar mistakes in the manuscript body and I am certain there are several ones I missed.

R: Thank you for your advice. English of the manuscript was revised by an agency as recommended by the journal. I hope the revised manuscript will meet the requirement of the journal. I also appreciate your efforts to point out many misuses of the language.

Some specific comments:

L050 – L052 could be reworded. Grammar seems off.

R: Thank you. We have revised.

L055 – L056 could be reworded.

R: Thank you. We have revised.

L083 Anthocyanins are the main pigments…

R: Thank you. We have revised.

L084 please clarify what anthocyanin characteristics.

R: Thank you. We have revised.

L085 “kind of unstable” is not accurate, are they less stable than other pigments? Please rephrase.

R: Thank you. We have revised.

L093 “light flowers”. I believe it should be lightly colored.

R: Thank you. We have revised.

L099 “is not only occurred” should be “does not only occur…”

R: Thank you. We have revised.

L109 there is no clear description of the objective of this study. This paragraph is a recollection of the methods used and not a concise experimental objective. Please reword.

R: Thank you. We have revised.

L157 needs rewording to improve conciseness.

R: Thank you. We have revised.

L160 – L162 needs better wording to improve clarity. Result presentation is confusing and a rewording is required. A better reference to Figure 2 in the text can aid in better describing the results.

R: Thank you. We have revised.

L442 “…chloride do not mean…” should be “…does not mean…”

R: Thank you. We have revised.

L557 Title has a typo. Method is written in present tense, it should be written in past tense.

R: Thank you. We have revised.

L573 Method is also written in present tense.

R: Thank you. We have revised.

L615 The conclusion is a recollection of results; however, it does not highlight the importance of this work and the repercussions of the information conveyed in the paper. Rewording is needed in the conclusion section to improve the overall quality of the paper.

R: Thank you. We have revised.

Overall, the manuscript's English quality is not adequate for publication. Thorough revision and corrections should be conducted before considering this paper for publication at IJMS. There are several typos and grammar mistakes in the manuscript body and I am certain there are several ones I missed.

R: Thank you. English of the manuscript was revised by an agency as recommended by the journal. I hope the revised manuscript will meet the requirement of the journal.
